# Is Repetitive Workload a Risk Factor for Upper Extremity Musculoskeletal Disorders in Surgical Device Mechanics? A Cross-Sectional Analysis

**DOI:** 10.3390/ijerph17041383

**Published:** 2020-02-21

**Authors:** Oliver Lotter, Tobias Lieb, Viktor Breul, Jochen Molsner

**Affiliations:** 1Department of Plastic, Aesthetic, Hand and Reconstructive Surgery, Academic District Hospital, Zeppelinstrasse 21, 78532 Tuttlingen, Germany; 2Office for Occupational and Hand Therapy, Neuhauser Strasse 85, 78532 Tuttlingen, Germany; tobiaslieb@gmx.de; 3Department of Medical Scientific Affairs, Aesculap AG, Am Aesculap Platz, 78532 Tuttlingen, Germany; viktor.breul@aesculap.de; 4IAS-Group for Occupational Health Management, Koenigstrasse 6, 78532 Tuttlingen, Germany; Jochen.Molsner@ias-gruppe.de

**Keywords:** work-related musculoskeletal disorders (WMSDs), upper extremity, repetitive work, surgical device mechanics, DASH score, Purdue Pegboard Test

## Abstract

To assess the prevalence of upper extremity work-related musculoskeletal disorders (WMSDs) among surgical device mechanics compared to a control group, a total of 70 employees were included and assigned to three occupational groups (grinders, packers, and control). Personal factors, work exposure, manual skill, and complaints were assessed by two self-administered questionnaires and an industry test. WMSDs were diagnosed in a standardised clinical examination. The two-one-sided *t*-tests (TOST) procedure was used to test the clinical equivalence of the respective grinding and packaging groups vs. the control group in terms of the Disabilities of the Arm, Shoulder and Hand (DASH) score. Thirty-nine study participants (56%) experienced at least one WMSD at the elbow, forearm, and/or wrist, mainly with signs of epicondylitis and nerve entrapment at the medial elbow. The risk of grinders developing upper extremity WMSD was about 2.5-times higher and packers had an 8.6-fold higher risk of a clinically relevant DASH > 29 compared to the control group. However, these differences were not statistically significant. The groups were also proven to be clinically equivalent in terms of DASH score. Surgical device mechanics do not seem to have worse DASH values or be at higher risk of upper limb WMSDs compared to a control group. This is the first study to analyse and compare different workplaces in this industry that are also common in other industries.

## 1. Introduction

Work-related musculoskeletal disorders (WMSDs) are pathologies in which the work environment and performance of work contribute significantly to the condition and/or the condition is made worse or persists longer due to work [1]. These disorders develop in the musculoskeletal system over a prolonged period of time and may limit activities in the professional environment or non-professional activities [2,3]. WMSDs describe a wide range of inflammatory and degenerative diseases that result in pain and functional impairment [4]. They are a significant occupational health problem among industrial and clerical workers with a significant medical, economic, and social impact in terms of absence due to sickness, cost of medical care, lost production, and personal suffering [5]. However, determining the true burden of morbidity associated with work activities still remains difficult within current surveillance and medical insurance systems [6]

Upper extremity pathologies related to work were first described in the 18th century by the Italian doctor and philosopher Bernardini Ramazzini, the father of occupational medicine. Today, the overall prevalence of upper extremity WMSDs ranges from 4% to 20% [7]. It has been estimated that the total cost of these disorders is between 0.5% and 2.0% of the gross national product in Scandinavian countries [8]. 

A PubMed search using the search terms “musculoskeletal disorder” AND “work” AND “upper extremity” (date of search: 12 July 2019) returned 1504 hits; from these hits 262 articles were shortlisted. Of 178/262 (68%) original research articles found, 60/178 (34%) publications contained a clinical examination of the test subjects with or without structured questionnaires/interviews. Of these, 50 were publications with a cross-sectional study design, but only 11 articles contained a control group.

Sorgaz described a repetitive strain injury model according to which highly frequent repetitive movements cause microlesions that accumulate in the affected musculoskeletal structures and eventually cause symptoms [9]. Epidemiological investigation indicates that adverse ergonomic exposures, vibration, forceful exertions, awkward postures, distribution of recovery periods, and duration of exposure are risk factors for developing WMSDs in the upper extremity [10,11]. However, there is conflicting information regarding monotonous and repetitive movements [12,13,14,15]. A multifactorial genesis of symptoms, including environmental, sociocultural, and personal characteristics, seems to be the leading hypothesis nowadays [16,17].

Upper extremity WMSDs have been shown to be related to numerous occupational situations and specific industries [18]. The medical device industry has not yet been analysed in this context. The objective of this study was to evaluate the prevalence of upper extremity WMSDs among surgical device mechanics compared to a control group. We hypothesised that disorders would not differ to a clinically relevant extent between surgical device mechanics performing specific repetitive work with and without forceful exertions compared to a control group.

## 2. Materials and Methods

### 2.1. Study Population: Subjects, Design, and Procedure

This study was conducted at the headquarters and main production site of Aesculap AG, Tuttlingen, Germany, which has approximately 3500 workers and employees. It is the world leading manufacturer of surgical instruments situated in a local area with more than 400 medical technology companies of varying sizes. Following an analysis of the workstations and work content, we defined three different study groups: group I = grinding and polishing characterised by repetitive and forceful exertions (subsequently referred to as “grinding”); group II = inspection and packaging characterised by repetitive exertions without force (subsequently referred to as “packaging”); and group III = all other white-collar and blue-collar employees working in manufacturing, warehouse, and office, representing a cross-section of the company not including groups I and II (but including other surgical device mechanics whose work does not include the activities performed by the first two groups) as a control group (hereinafter referred to as “control”). A detailed description of the job tasks is listed in Table 1. Representative pictures of the workstations are shown in Figure 1 and Figure 2.

**Control:** The control group represents a cross-section of the company’s working population. All workplaces at the company were included, from light office work to heavy work with high physical strain.

The study was approved by the Ethics Committee of the State Medical Council of Baden-Wuerttemberg, Jahnstrasse 40, 70,597 Stuttgart, Germany (project number F-2017-005). Random samples of active white-collar and blue-collar workers from the targeted groups were taken via randomisation lists between September 2017 and March 2018 and this sample population was asked to fill in two self-administered questionnaires after having agreed to enter the study and signed the informed consent form. No incentives were offered. The first standardised questionnaire obtained information about exclusion criteria (see below), demographic, and personal data, such as sex, handedness, secondary occupation, sporting and physical hobbies (categorical parameters), age, height and weight (body mass index), volume of employment, and years of service (continuous parameters). Current subjective complaints in the upper extremities (i.e., symptoms) were also recorded in this questionnaire. The validated Disabilities of the Arm, Shoulder and Hand (DASH) outcome measure was used as a second questionnaire [19,20]. The results of this 30-item questionnaire were used to calculate a scale score ranging from 0 (no disability) to 100 (most severe disability), known as the disability/symptom score. In addition, data was also collected using two optional modules intended to measure symptoms and function in athletes, performing artists, and other workers, whose jobs require a high degree of physical performance (sport/music and work scores).

The following exclusion criteria were applied in this study: Age <18 or >65 years.Employment in the respective workplace for less than 5 years.Currently on sick leave.Absence from work due to upper extremity pain for more than 2 weeks within the last 3 months.Cervical spine syndrome or herniated intervertebral disc.Shoulder pain radiating into the forearm.Debilitating congenital malformation of the upper extremity.Rheumatoid conditions including fibromyalgia.Previous upper extremity surgery due to nerve entrapment syndrome(s) and/or chronic musculoskeletal disorders, such as tennis elbow, golfer’s elbow, tenosynovitis of the flexor and/or extensor tendons, including trigger finger and de Quervain’s disease.More than three unanswered items in the DASH disability/symptom questionnaire.

### 2.2. Objectives

We applied the DASH to measure physical function and symptoms in patients with one or more musculoskeletal disorders in the upper limb [21]. We chose an equivalence design with a margin of 15 points to assess the clinically detectable differences between different occupational groups [22]. We also collected data on secondary endpoints in order to evaluate the outcome of the exposure (see clinical data and Purdue Pegboard Test).

### 2.3. Clinical Data

The participants were invited for a structured physical examination of the elbow, forearm, wrist, and hand. It consisted of active range of motion (ROM) manoeuvres of the wrist (as measured in 3 planes using the neutral zero method), grip strength by Jamar dynamometer (3 measurements per side), pain on a visual analogue scale (VAS) at rest and under strain, as well as a search for tender points for tenosynovitis. The examiner decided the diagnoses based on a standard set of criteria for pathognomonic clinical signs specific to upper extremity pathologies as proposed by Waris et al. [23]. These included pain on the radial side of the wrist together with Finkelstein’s test for de Quervain’s disease, lateral epicondyle pain and Maudsley’s test for tennis elbow, and the combination of Hoffman–Tinel sign and static 2-point discrimination (2-PD) for finger sensibility for nerve entrapment syndromes, adding Phalen’s test specific for carpal tunnel syndrome [24,25,26,27,28]. 

### 2.4. Purdue Pegboard Test 

The participants also underwent the Purdue Pegboard (PPB) Test, which is a neuropsychological test that involves different abilities: gross movements of arms, hands, and fingers, as well as fine motor extremity (also known as “fingerprint” dexterity), and bimanual dexterity [29,30]. It was originally intended as an industrial test for assessing the dexterity of assembly line workers [31]. Nowadays, the PPB Test is used, for example, to follow-up on and assess disabilities and limitations. The pegboard consists of a board with two parallel rows of 25 holes each into which cylindrical metal pegs are placed by the examinee. The test involves a total of four trials [32]. The subsets for preferred, non-preferred, and both hands require the patient to place the pins in the holes as quickly as possible, with the score being the number of pins placed in 30 s.

There were only two different examiners (one for physical examination and one for the PPB Test) to optimise inter-rater reliability. The examiners were blinded to the questionnaire responses.

### 2.5. Statistical Analysis 

All statistical analyses were performed using the SAS software version 9.4 (SAS Institute Inc./Cary, NC, USA). The primary analysis consisted of two parallel tests, which proved the hypothesis of equivalence of a respective test group and the reference group. This confirmatory analysis referred to the disability/symptom DASH score (without the two optional modules) and used the two-one-sided *t*-test (TOST) procedure [33]. Both the grinding and the packaging groups were compared to the control group. The significance levels of the equivalence tests were adjusted according to Bonferroni, resulting in two-sided *t*-test levels of 0.025 [34]. The equivalence of two groups was considered proven if the observed difference between them was significantly lower than the equivalence margin, which was set to 15 points [35]. The 95% simultaneous confidence intervals for differences from the control (Dunnett’s method) were used to compare DASH scores [36].

For descriptive statistics with approximately normally distributed continuous data, the mean value and standard deviation (SD, shown in parentheses) were calculated. For clearly non-normally distributed data, especially skewed continuous data, the median and interquartile ranges (IQR, shown in parentheses with “-” between the two values) were calculated [37]. For categorical variables, absolute and relative frequencies were calculated. When reporting percentages for categorical data, the numerators and denominators of the calculations are always given in parenthesis. 

Where statistical tests have been used for comparisons between groups, the two-sided p-values must be considered as measures of the effect size rather than as confirmatory values. 

The sample size was calculated using nQuery Advisor 7.0 (Statistical Solutions Ltd., Cork, Ireland). The one-sided significance level used for individual *t*-tests in the TOST was set to 0.013. Assuming an expected group mean difference of 0 and a pooled standard deviation of 14.68, the study has a power of 80% to detect equivalence when the sample size is 26 per group [38]. 

## 3. Results

Based on a basic population of about 3500 workers and employees, the total population after application of the inclusion criteria (age between 18 and 65 years, employment in the respective workplace for more than 5 years and not currently on sick leave) consisted of 63 subjects in group I, 208 in group II, and 2501 in group III (Figure 3). A random selection was made from this population for inclusion in the study. After completion of the questionnaires and taking the exclusion criteria into account, a total of 70 subjects (grinding *n* = 20, packaging *n* = 24, and control *n* = 26) were included in the study. Six subjects were excluded, five for medical criteria and one because of absence from work due to upper extremity pain for more than two weeks within the last three months.

### 3.1. Demographic Data

The majority of the study population was male (67% (47/70)) and right-handed (83% (55/70)). Very few had a secondary occupation (9% (6/70)) and 61% (43/70) reported having sporting or physical hobbies. The three groups had comparable demographic data with regard to age (42.1 (12.2) years), body mass index (BMI) (26.2 (5.0) kg/m^2^), full volume of employment (91% (64/70)) and years of service at the company (16.1 (9 to 28) years) (Table 2 and Table 3).

### 3.2. DASH Outcome Measure

The DASH scores for each of the three groups, including the optional sports/music and work modules, are shown in Table 4. Our mean scores are in good agreement with the normative DASH scores [38]. Remarkably, the values from the grinding and packaging groups were about double those the control group when we analysed the sports/music and work modules. Note that lower DASH scores are associated with a better situation. 

### 3.3. Clinical Relevance Testing

The distribution of the values of the DASH score in the three groups is shown in Figure 4. Both separate TOST-based tests for equivalence (grinding vs. control and packaging vs. control) demonstrated equivalence with the 15-point margin (all *p*-values < 0.0001). The mean score differences with corresponding 95% confidence limits are shown in Table 5.

The primary analysis proved that both grinding and packaging were equivalent to the control group. As the equivalence margins were set in respect to the minimal clinically important difference (MCID), these results support the evidence that the differences between the groups were not clinically relevant. Aside from the clinical relevance, the observed differences were proven to be statistically significant with 95% confidence intervals (Table 6). 

As both 95% simultaneous confidence intervals for difference do contain zero, the significance of the observed differences at the 5% level can also be denied. 

Exploratively, we divided the DASH score in the individual groups according to gender. A clearly different structure in the distribution between men (homogeneous) and women (variable) was observed (Figure 5).

### 3.4. Clinical Data

Upper extremity pain (i.e., symptoms) was reported in the questionnaire by 40% (8/20) of grinders, 58% (14/24) packers, and 42% (11/26) of the control group. 

Pathognomonic clinical signs for upper extremity WMSDs (i.e., diagnoses) at the elbow, forearm and/or wrist (trigger finger, Finkelstein’s test, Maudsley’s test, Hoffman–Tinel sign, and Phalen’s test) were found in 56% (39/70) of the people examined. The frequency of one or more diagnoses was 60% (12/20) for grinders, 58% (14/24) for packers, and 50% (13/26) in the control group (Table 7). Bilateral manifestation was present in 34% (24/70) and 14% (10/70) of the subjects had two or more different pathologies in the ipsilateral limb. In the case of a positive Hoffmann–Tinel sign, it was mostly present at the medial elbow as a sign of ulnar tunnel syndrome. 

The sensitivity of the upper extremity pain (symptoms) to the gold standard of pathognomonic clinical signs (diagnoses) was 54% (21/39) whereas the specificity was 61% (19/31) (Table 8). 

In our sample there were three packers with a DASH score >29, while in the control group there was no such case (OR 8.63 (0.40; 186.88)). In the grinder group, 14 subjects were diagnosed with upper extremity WMSD, in the control group these were only 12 (OR 2.59 (0.76; 8.78)). Further odds ratios are shown in Table 9.

Continuous clinical parameters are shown in Table 10. A visual analogue scale (VAS) was used for pain evaluation at rest and under stress. Wrist mobility (range of motion in three planes) and grip strength were also collected. No relevant differences could be found between the groups. We did not expect any relevant differences between the right and left hand, therefore we used mean values for further analysis. When comparing grip strength with reference values from a healthy population, subdivided according to sex and age group, below-average values were found in all three groups [39]. The values for men were twice as high as the values for women (Figure 6). 

### 3.5. Purdue Pegboard Test

The results of the Purdue Pegboard Test (PPB) are shown in Table 11. Test 4 was chosen to represent the PPB as this summarises Tests 1 to 3 well by adding them up. As Tests 1 to 3 may be expected to correlate, Test 4 should show the potential differences between the groups more clearly. The intergroup comparison of the Purdue Pegboard Test showed no relevant differences. When comparing the values with the reference values, the values were mostly below average, e.g., 44.2 points for men and 41.2 points for women for the PPB Test 4 [40].

## 4. Discussion

Upper extremity WMSDs represent a range of disorders of the muscle, tendon, or nerve that also can occur in non-workplace settings with a similar or identical pathophysiology [16,41]. Musculoskeletal disorders are the most expensive disease category with regard to work absenteeism and disablement. We chose to survey a population of actively employed surgical device mechanics and compared them with a group of employees believed not to be exposed to repetitive hand and arm movements to such a large extent.

We conducted this prospective study at the headquarters and main production site of Aesculap AG, which is a leading manufacturer and global player in the surgical instrument industry, located in a medical technology cluster region [42,43]. Among other things, the company is valued for its comprehensive preventive medical treatment programmes that are easy for all its employees to access. One of the basic requirements for our study was for many of the employees to have been with the company for a long time and to have had a constant workplace over a long period of time. 

### 4.1. DASH and Other Measurement Values

The DASH outcome measure has proven to be a reliable and valid instrument to measure physical function and symptoms in upper extremity musculoskeletal complaints, including WMSDs [44,45]. It is the best instrument for evaluating patients with upper limb joint disorders [46]. 

As in previous studies, it was observed that a non-clinical “normal” population does not have a zero value [47]. Both groups of manual workers tested in the current study proved their statistical equivalence with the control group. Since our equivalence margins correspond to the minimal clinically important difference (MCID) for the DASH score, we were able to state that neither grinders nor packers were clinically different from the control group. The mean DASH score in our investigation was 8.5 (SD 7.6) points for the grinder group, 12.0 (SD 10.6) points for the packer group, and 7.9 (SD 8.1) points for the control group. Although the control group had the best (=lowest DASH) score, the differences were not statistically significant. The reference value of 10.1 (SD 14.7) points was only exceeded by the packers; the other two groups were better than this threshold. 

Surprisingly, it is not the grinders with their loading and repetitive activity tasks that have the highest (=worst) DASH score, but the packers whose work tasks are defined as repetitive with less loading and more variability in the movement sequences. This finding contrasts with a study which has shown significant differences in DASH scores between manually demanding and manually low-demanding jobs in favour of the latter [48]. One reason for this could be that working hours and job designs have changed significantly in the medical device industry in recent decades so there may be no relevant differences to other occupational groups with a lower workload. A lot of grinders repeatedly carry out pre-processing and post-processing activities, which lead the grinder away from the whetstone during a shift at today’s modern workplaces. This is not the case with packers who mainly carry out the same work throughout a shift. A further explanation could be that the grinders are usually very experienced specialists who would be hard to replace in the event of health problems. This means that there is a particularly high level of motivation on the part of employers for health protection measures in this group. According to common doctrine, static work processes favour the development of upper extremity WRMDs, which should represent a risk for grinders [49]. In reality, however, there is no difference from the somewhat more variable and multidimensional movement sequences of the packers. It may even be possible that microvibrations in grinders could have a prophylactic effect preventing the development of WRMDs in the upper extremity [50]. 

The average values of the three investigated groups correspond to those of a population not employed in manual work. A literature review showed that our scores are generally lower than in most studies that have analysed normative values in the general population outside clinical settings [37,48,51,52,53]. We gained the impression that some test subjects had previously reported clinical complaints well below a DASH score of 29, which was defined by Williams as a kind of threshold [54]. In our sample, the risk of packers having clinically relevant complaints (DASH invalidity/symptom score >29) was higher (odds ratio 8.63 (0.40; 186.88)) than in the control group. However, this did not reach statistical significance.

Interestingly, the DASH showed a homogeneous distribution between the three groups in men, while women had a very inhomogeneous distribution. The influence of age (increasing DASH scores with increasing age) and gender (higher DASH scores in women) as well as further co-variates in this context will be topic of our future studies [48,52].

Active range of motion of the wrist and pain on the visual analogue scale (VAS) showed no abnormalities or relevant differences between the groups. The measured values for grip strength were partly below the published reference values, taking into account handedness, age, and sex [38,48]. In the Purdue Pegboard Tests our subjects mostly performed below than the reference values given [40,55]. This result is astonishing, because we would have expected a different result, especially for packers’ repetitive and fine motor activities as well as for those of numerous workplaces in the inspection department.

### 4.2. Prevalence of Symptoms and Diagnoses

The design of our cross-sectional study with a control group, physical examination of the subjects, and the use of a questionnaire can be found in several studies related to different industries. Our literature research showed a prevalence of upper extremity pain (i.e., symptoms) of between 21% and 71% in the study group and between 6% and 50% in the control group [56,57,58,59,60,61,62,63,64,65,66,67,68]. This is in line with our investigation where symptoms were reported in the first questionnaire by 8/20 (40%) grinders, 14/24 (58%) packers, and 11/26 (42%) people in the control group. 

In most studies, only the symptoms of musculoskeletal disorders of the upper extremity were described, instead of additionally identifying pathognomonic clinical signs (i.e., diagnoses). In the latter the prevalence has ranged from 21% to 56% compared to 5% to 22% in the control group [57,58,60,61,62,64]. Upper limb musculoskeletal disorders were diagnosed in 14/20 (70%) grinders, 13/24 (54%) packers, and 11/26 (42%) of the test persons in the control group for our investigation, with bilateral manifestation in approximately one third of cases. Thus, the frequency of reported symptoms is lower than the frequency of detected musculoskeletal disorders in our study, in contrast to previous studies which showed a high prevalence of complaints but a relatively low number of definite diagnoses [69,70]. One explanation could be the rather low sensitivity of the questionnaires to upper extremity complaints (42%–65%), as already found by Ohlsson et al., which underlines the need for a combination of questionnaire and clinical investigation in such cross-sectional studies [71]. Another aspect could be the focus on diseases of the elbow, forearm, wrist, and hand in our investigation, which did not consider more proximal pathologies of the upper extremity, such as shoulder disorders. The latter are sometimes unspecific and there are hardly any pathognomonic clinical tests to make a reliable diagnosis. This could, in turn, lead to a gap between symptoms and diagnoses of diseases.

It has been shown that musculoskeletal disorders in different body regions are associated with different branches of industry. Lateral epicondylitis, wrist tendinitis and carpal tunnel syndrome are considered the most common diseases [57,60,61,62,64,70]. In our investigation, 30% of all test persons and 54% of the subjects with upper extremity complaints had signs of medial epicondylitis (golfer’s elbow) and nerve entrapment at the medial elbow (cubital tunnel syndrome), followed by lateral epicondylitis (tennis elbow) in 14% and 27%, respectively. Our literature research revealed a prevalence of medial epicondylitis (golfer’s elbow) of 3% to 20% in occupational settings associated with forceful activities but without clear reference to repetitive work [64,72,73,74,75]. Medial epicondylitis occurs in only 10%–20% of all epicondylitis as a result of common flexor tendon microtrauma and degeneration, affects men and women equally, and is often associated with cubital tunnel syndrome [76]. We were able to confirm this above-average prevalence and the frequent co-occurrence of the two pathologies in our study. The risk for grinders of developing upper extremity WMSD was higher than in the control group (odds ratio 2.59 (0.76; 8.78)). However, there was no statistically significant increased occurrence of these pathologies in any of the groups.

### 4.3. Interpretation of the Results

We hypothesised that there would be no clinically relevant differences in the prevalence of upper extremity WMSDs between surgical device mechanics with specific repetitive work tasks, with and without forceful exertions, compared to other employees in the same industry. We were able to confirm this with surgical device mechanics not seeming to be at higher risk of upper limb WMSDs compared to the control group. Although grinders and packers had higher DASH values than the control group, this difference was of no clinical relevance and statistically not significant. However, there are partly contradictory statements in this regard from other sectors of industry [12,13,14,15,76]. The reasons for our findings are probably that the exposed groups did not vary much in terms of their physical workload and repetitive work tasks. However, since WMSDs represent a complex condition that can be influenced by individual, environmental, psychosocial, and organizational factors, this simplified interpretation is only one possible aspect in a multi-level approach with numerous interactions and thus might explain some contradictory results [77,78]. Variations in the duration of exposure, work interruptions through less monotonous and stressful pre-processing and post-processing activities, psychosocial factors, the level of training, and income disparities may have such an influence that the primary work tasks lose relevance with regard to upper extremity WMSDs [65,66].

The diagnostic patterns observed in this investigation were, in general, consistent with medial epicondylitis and cubital tunnel syndrome in the elbow. Lateral epicondylitis, tenosynovitis in the wrist, and carpal tunnel syndrome were underrepresented. This can also be explained by the working positions at various workplaces, some of which were noticeable during our site inspections, with partially unpadded support of the elbows and proximal forearms on edges and surfaces (Figure 7). 

In addition to regular occupational health check-ups, the study company offers optional preventive medical check-ups for individual complaints. Furthermore, there are check-up programs for each employee on a voluntary basis to identify risk factors, preliminary stages of illness or diseases, and to be able to prevent or treat them in a targeted manner. In the latter case, a full-body profile is created including patient history, blood values, Electrocardiogram (ECG), lung function testing, visual and hearing tests, as well as a full body examination, including different organ systems. If particularities are identified during these examinations, further targeted measures may be taken. In the case of upper extremity WMSDs, hand therapeutic measures, ergonomic optimization of the workplace, and examinations by specialists (e.g., upper extremity surgeon) are usually initiated. Based on our study results, the response to the strikingly high rates of medial elbow pathologies should be an ergonomic sitting position and simple padding of the table edges. Longer periods of repetitive work, especially for packers, should be interrupted by other types of work, including job rotation as a preventive measure.

### 4.4. Limitations of Our Investigation

There is a possible evaluation bias as it was not possible to blind clinical evaluators due to the participants’ clothing (blue collar vs. white collar workers). Thus, we believe that the bias is small, if present. A selection bias is also likely as persons with upper extremity complaints could be more motivated to volunteer. However, since this effect applies equally to the study groups and the control group, it would be cancelled out and at the same time increase the total number of complaints. This last effect may partly be compensated by the healthy worker effect, which often results in the working population being statistically shown to be healthier than is actually the case. People with pain and functional disorders might leave their work, and cross-sectional studies thus may underestimate the risk. The only validated test available for our series is the self-reported DASH outcome measure. However, the ability of subjects with musculoskeletal disorders cannot be assessed on the basis of questionnaires alone. As in every questionnaire, the translation of scores from the questionnaire to functional capacity for work can be questioned. Furthermore, our cross-sectional data has inherent limitations, including the inability to clearly distinguish clinically between the chronic, recurrent, and acute symptoms we observed. In addition to biomechanical constraints, psychosocial factors with difficult-to-define effects have been shown to play a role in the genesis of WMSDs [79,80,81]. We have not included this aspect as a contributing cause in our analysis to prioritise other questions and due of the high complexity of its evaluation. In fact, grinders are skilled workers with good pay, while packers are mostly lateral entrants and semi-skilled workers with lower incomes. As we performed a cross-sectional study, the causality of our reported pathologies could not be assessed. We could not adjust for workers’ baseline risk of WMSDs resulting from prior occupational exposures or other non-occupational exposures. In this case, a longitudinal study, including the time dimension, would be advantageous in order to better decide what changes should be made to working conditions. The overall very high level of preventive measures and occupational health care for the entire workforce could be a reason why no significant differences could be found between the groups. Despite the size of the company, the subdivision into specific working groups with certain tasks leads to a relatively small number of cases, which is especially true for the grinders. However, a smaller subdivision would have reduced the selectivity. The sample size was calculated based on anticipated effect in order to enable the proof of the study hypothesis of equivalence between occupational groups (see statistical analysis). The targeted sample size could not be achieved in all groups due to lack of consent in smaller subpopulations. Nevertheless, the actual sample sizes were sufficient to prove the study hypotheses.

Planning the current study, we assumed a normal distribution for the DASH score. The statistical analyses followed the a priori plan in order to maintain the significance level. The distribution analysis of the study data revealed an obvious ceiling effect, with 12 of 70 participants having the best score of 0 pts. Normal distribution-based statistical methods may lose sensitivity in terms of detecting differences between groups, with respect to the power to detect effects in such a situation. The ceiling is not unusual in studies using scores for outcome evaluation, particularly studies in orthopaedics, and should be taken into account when planning statistical analysis for future studies using DASH scores [82].

### 4.5. Strengths of Our Investigation 

The strengths of this study include the comparatively homogeneous study groups of healthy and physically active subjects as well as the above-average length of service of the employees with low job turnover, which should keep confounding to a minimum. In a lot of studies of WMSDs, the workplace data come from case series compiled from records of reportable injuries and illnesses kept by the employer’s liability insurance companies and the in-plant medical departments. Since these sources identify late-stage disorders that prompt workers to seek medical attention, the data probably identify an incomplete subset of cases and underestimate the true rates of disease. With regard to the study design of our investigation, the combination of self-administered questionnaire and specific face-to-face physical assessment has proven effective in obtaining a consistent study result. Thereby the criteria for specific and reliable assessment of the functional capacity of the forearm and hand according to Wind et al. were met [44]. Furthermore, the physical examinations were performed by the same hand surgeon and the testing by the same hand therapist. Such a set-up was very rare in previous studies and may have influenced the frequency of detected diagnoses of WMSDs. Our aim was to keep well-defined disorders separate from more diffuse conditions. When examining the test subjects, we adhered to the classification system as described by Van Eerd et al., which proposes specific diagnoses for muscle, tendon, and nerve disorders rather than a description of the signs and symptoms found [45]. 

## 5. Conclusions

Our findings do not indicate that repetitive work with and without forceful exertions is associated with a higher risk of upper extremity WMSD in surgical device mechanics compared to a control group, but small effects cannot be ruled out. Although a high standard of occupational health and safety measures can be assumed within the study company, the pathologies frequently found in the medial elbow may be due to specific working practices. Based on the results of this study, targeted occupational health measures can be designed and implemented with the aim of both preventing and treating upper extremity musculoskeletal disorders [83]. Once these measures have been implemented, we recommend re-evaluating the groups over time in a longitudinal study. This is the first study to analyse this specific industry and compare different workplaces and their work tasks. Comparable loading and repetitive activities can also be found in other industries.

## Figures and Tables

**Figure 1 ijerph-17-01383-f001:**
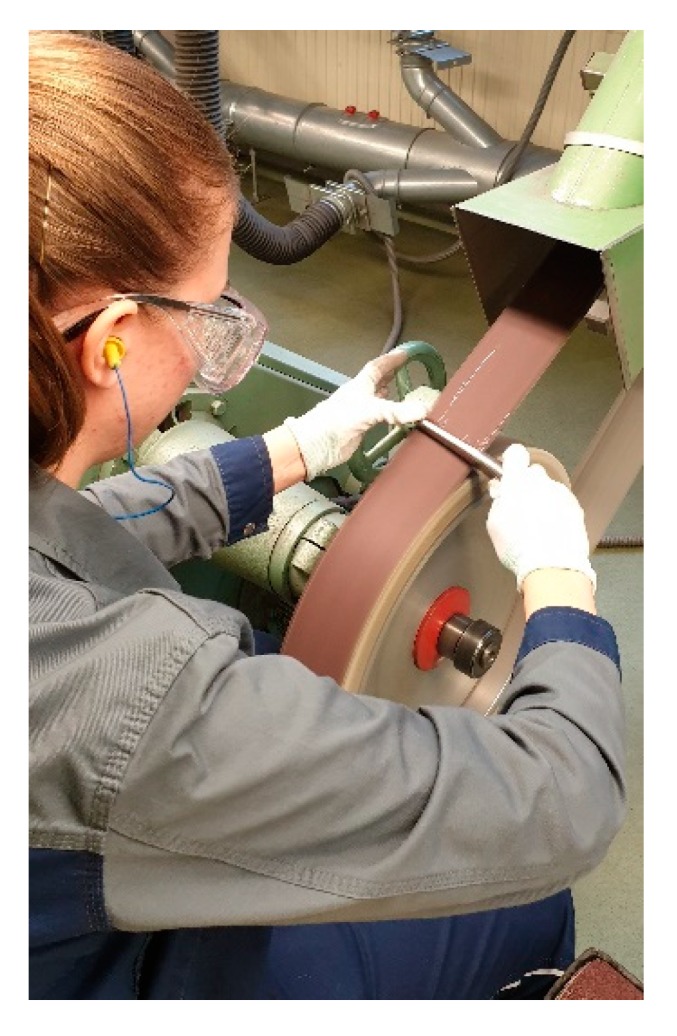
Workplace grinding.

**Figure 2 ijerph-17-01383-f002:**
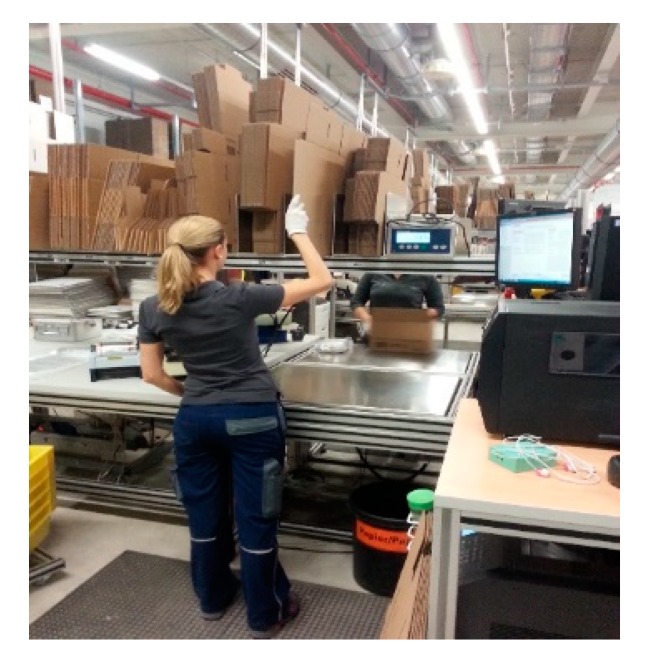
Workplace packaging.

**Figure 3 ijerph-17-01383-f003:**
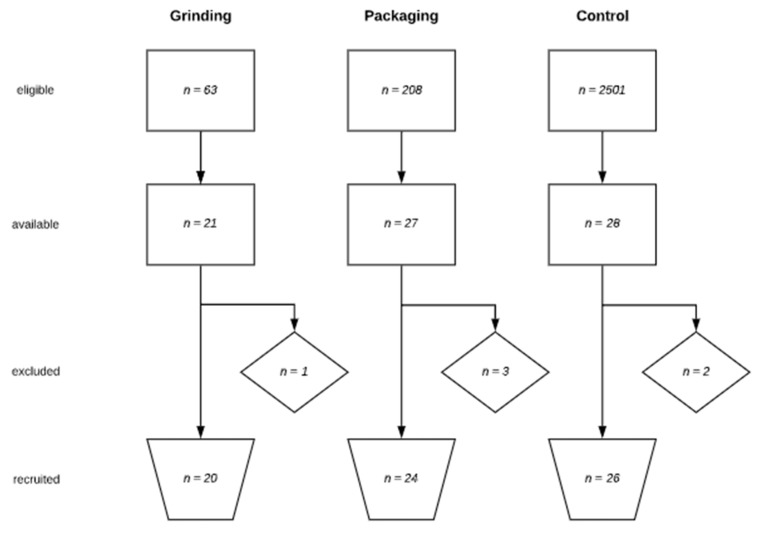
Study population flowchart.

**Figure 4 ijerph-17-01383-f004:**
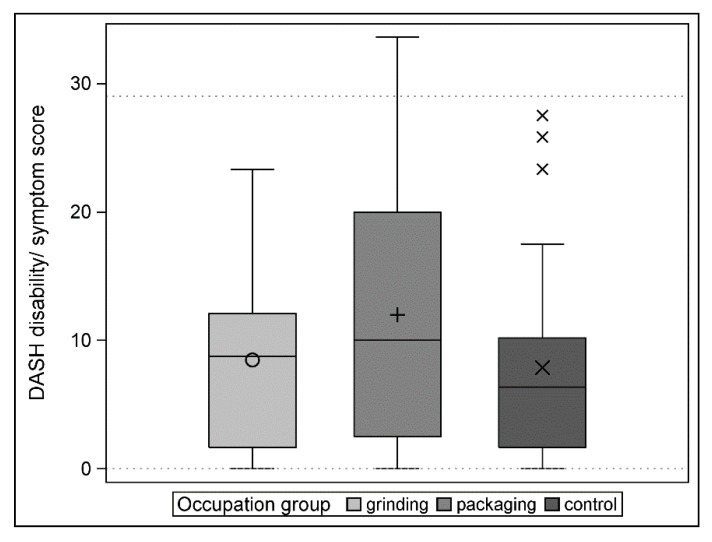
Distribution of the DASH score in the three groups.

**Figure 5 ijerph-17-01383-f005:**
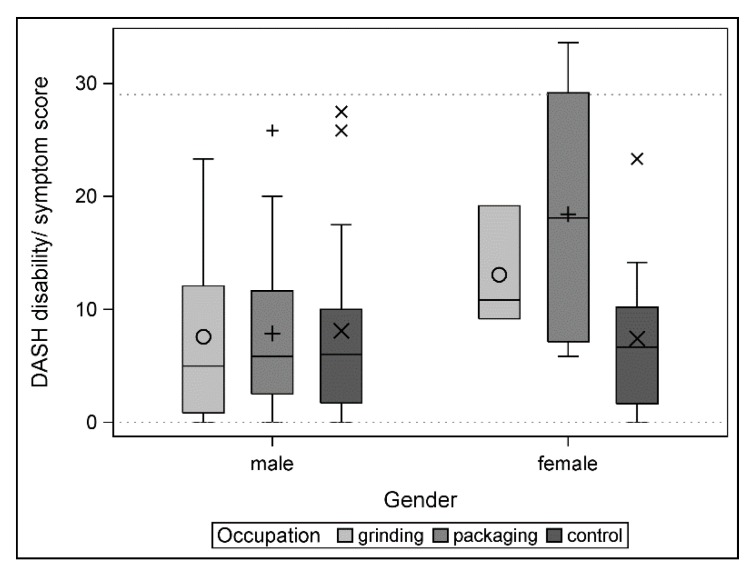
Distribution of the DASH disability/symptom score by gender.

**Figure 6 ijerph-17-01383-f006:**
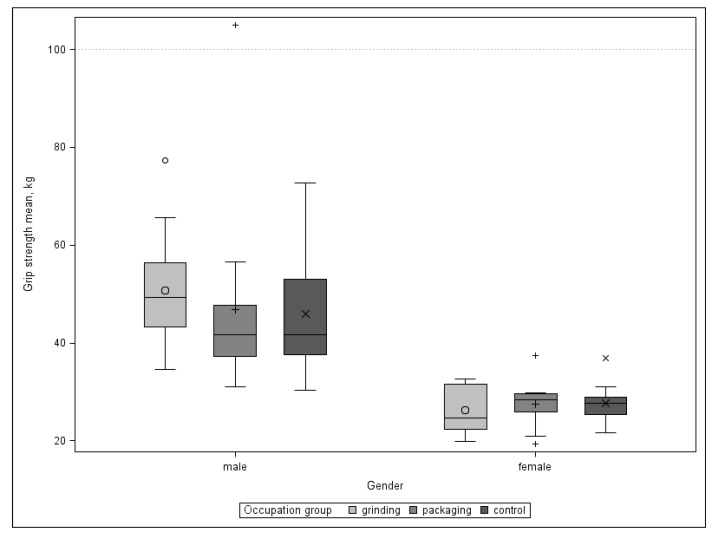
Grip strength by gender.

**Figure 7 ijerph-17-01383-f007:**
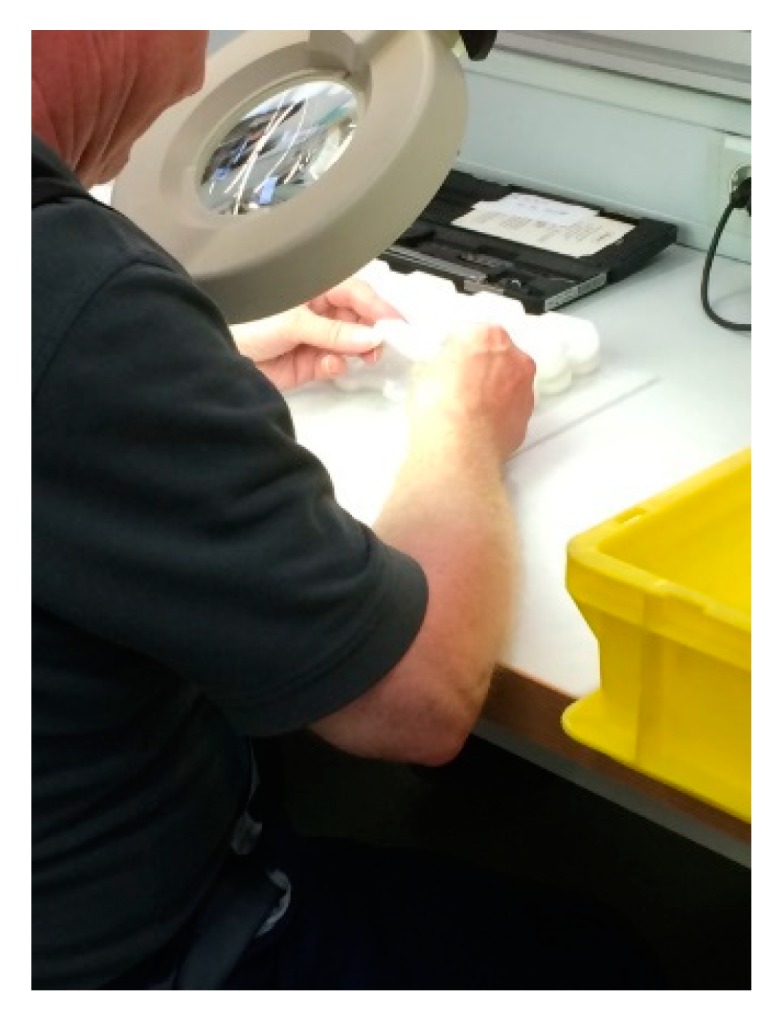
Unpadded position of the elbow and forearm creating risk of medial epicondylitis and ulnar nerve compression.

**Table 1 ijerph-17-01383-t001:** Description of job tasks.

	Grinding	Packaging
Work contents	Grinding and polishing of workpieces	Packaging of products
	Pre- and post-processing (acceptance of orders, control, final cleaning)	Pre- and post-processing (acceptance of orders, inspection)
Main activity time proportion	60%–80%	80%–90%
Working position	Sitting	Both sitting and standing depending on the workstation
Weight of product/packaging	1 kg, on average significantly less, higher total load	10 kg, on average significantly less, lower total load
Use of hands	The leading hand is not changed (one-sided static load)	Both hands (all planes of motion with high ROM)
Range of motion	Low	High
Load	Medium-heavy	Light to medium-heavy

ROM = range of motion.

**Table 2 ijerph-17-01383-t002:** Demographic data: categorical parameters.

	Grinding (*n* = 20)	Packaging (*n* = 24)	Control (*n* = 26)
Sex			
Female	25% (5/20)	38% (9/24)	35% (9/26)
Male	75% (15/20)	62% (15/24)	65% (17/26)
Handedness			
Right	70% (14/20)	92% (22/24)	85% (22/26)
Left	30% (6/20)	8% (2/24)	15% (4/26)
Secondary occupation	15% (3/20)	4% (1/24)	8% (2/26)
Sporting and physical hobbies	55% (11/20)	54% (13/24)	73% (19/26)
Employment volume less than full time	5% (1/20)	13% (3/24)	8% (2/26)

**Table 3 ijerph-17-01383-t003:** Demographic data: continuous parameters.

	Grinding (*n* = 20)	Packaging (*n* = 24)	Control (*n* = 26)
	Mean (SD)/Median (IQR)	Mean (SD)/Median (IQR)	Mean (SD)/Median (IQR)
Age (years)	41.7 (13.1)	42.6 (11.9)	42.0 (12.3)
BMI (kg/m^2^)	27.1 (4.6)	25.3 (5.6)	26.4 (4.6)
Years of service	13.5 (7–31)	13.8 (9–28)	17.3 (10–27)

**Table 4 ijerph-17-01383-t004:** Disabilities of the Arm, Shoulder and Hand (DASH) scores.

	Grinding (*n* = 24)	Packaging (*n* = 24)	Control (*n* = 26)	Reference Values
	Mean (SD)	Mean (SD)	Mean (SD)	Mean (SD)
DASH score	8.5 (7.6)	12.0 (10.6)	7.9 (8.1)	10.1 (14.7)
DASH sports/music score	12.5 (13.9)	12.0 (18.2)	4.5 (8.0)	9.8 (22.7)
DASH work score	14.9 (15.3)	13.0 (18.1)	6.8 (10.5)	8.8 (18.4)

**Table 5 ijerph-17-01383-t005:** Results of the TOST-based tests.

Test	Mean Difference with 95% CI	Equivalence Range	Assessment
Grinding vs. control	0.6 (−4.3, 5.5)	(−15; 15)	Equivalent
Packaging vs. control	4.1 (−1.2, 9.5)	(−15; 15)	Equivalent

**Table 6 ijerph-17-01383-t006:** Results of Dunnett’s test.

Test	Mean Difference with Simultaneous 95% CI (Dunnett)	Assessment
Grinding vs. control	0.6 (−5.6, 6.8)	CI contains 0
Packaging vs. control	4.1 (−1.7, 9.9)	CI contains 0

**Table 7 ijerph-17-01383-t007:** Clinical data: prevalence of pathognomonic clinical signs.

	Grinding (*n* = 20)	Packaging (*n* = 24)	Control (*n* = 26)
Pain at elbow/forearm and/or wrist	40% (8/20)	58% (14/24)	42% (11/26)
Trigger finger	10% (2/10)	17% (4/24)	4% (1/26)
Finkelstein’s test	15% (3/20)	13% (3/24)	4% (1/26)
Maudsley’s test (middle finger test)	15% (3/20)	17% (4/24)	12% (3/26)
Hoffman–Tinel sign	40% (8/20)	21% (5/24)	31% (8/26)
Phalen’s test	20% (4/20)	8% (2/24)	4% (1/26)

**Table 8 ijerph-17-01383-t008:** Cross table with clinical signs (diagnoses) and upper extremity pain (symptoms).

		Upper Extremity Pain	
		No	Yes
Clinical signs	No	27% (19/70)	26% (18/70)
	Yes	17% (12/70)	30% (21/70)

**Table 9 ijerph-17-01383-t009:** Odds ratios for important outcomes.

Endpoint	Effect	Odds Ratio Estimate (Lower; Upper 95% Confidence Limit)
DASH > 29	Grinding vs. control	1.29 (0.02; 74.11)
DASH > 29	Packaging vs. control	8.63 (0.40; 186.88)
Arm pain	Grinding vs. control	0.92 (0.28; 2.99)
Arm pain	Packaging vs. control	1.86 (0.61; 5.72)
Any diagnosis	Grinding vs. control	2.59 (0.76; 8.78)
Any diagnosis	Packaging vs. control	1.36 (0.45; 4.14)

**Table 10 ijerph-17-01383-t010:** Clinical data: continuous parameters.

	Grinding (*n* = 20)	Packaging (*n* = 24)	Control (*n* = 26)	Reference Values
	Mean (SD)	Mean (SD)	Mean (SD)	Mean (SD)
VAS at rest (points)	1.4 (0.9)	2.0 (2.0)	1.5 (1.0)	-
VAS at stress (points)	2.5 (2.5)	3.0 (2.3)	2.2 (1.9)	-
ROM E/F (degrees)	125.6 (15.9)	125.5 (14.5)	123.8 (11.3)	-
ROM S/P (degrees)	176.0 (8.2)	177.8 (7.3)	178.8 (5.9)	-
ROM U/R (degrees)	50.5 (2.2)	51.4 (3.8)	51.0 (3.7)	-
Grip strength (kg)—male	50.7 (11.1)	46.8 (17.5)	46.0 (11.8)	54 (7)
Grip strength (kg)—female	26.2 (5.7)	27.5 (5.3)	27.8 (4.6)	32 (6)

VAS = pain on visual analogue scale, ROM = range of motion, E/F = extension/flexion, S/P = supination/pronation, U/R = ulnar/radial abduction.

**Table 11 ijerph-17-01383-t011:** Results of the Purdue Pegboard Test (in points).

	Grinding (*n* = 20)	Packaging (*n* = 24)	Control (*n* = 26)	Reference Values
	Mean (SD)	Mean (SD)	Mean (SD)	Mean (SD) *
Preferred hand (Test 1)	14.30 (2.08)	15.00 (2.09)	15.42 (2.18)	15.47 (1.8)
Non-preferred hand (Test 2)	13.60 (2.04)	14.25 (1.89)	14.62 (2.37)	14.94 (1.86)
Both hands (Test 3)	11.65 (2.18)	11.54 (1.25)	12.35 (1.62)	12.8 (1.98)
Right + left + both hands (Test 4)	39.55 (5.00)	40.79 (3.80)	42.38 (5.32)	43.21 (-)
Assemblies (Test 5)	29.95 (7.42)	30.17 (5.14)	33.54 (6.41)	38.19 (6.25)

* Calculated for the mean age category (40–49 years) and for the study population gender ratio (23/70 female) according to [40].

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
