# Peer review of "Is Repetitive Workload a Risk Factor for Upper Extremity Musculoskeletal Disorders in Surgical Device Mechanics? A Cross-Sectional Analysis"

_ijerph, 2020, doi:10.3390/ijerph17041383_

Round 1
Reviewer 1 Report
The article entitled “Is repetitive workload a risk factor for musculoskeletal disorders in surgical device mechanics? A cross-sectional analysis”: shows unpublished data regarding the incidence of upper extremity work-related musculoskeletal disorders (WMSDs) among surgical device mechanics. The article appear well designed and clinical data and statistical analysis well done.
However, I would like to make only one comment:
I don't understand the importance of including the "Literature research" point in materials and methods. I think the authors could analyze the previous literature in the introduction.
Author Response
Point 1: I don't understand the importance of including the "Literature research" point in materials and methods. I think the authors could analyze the previous literature in the introduction.
Response 1: The literature research was removed from the chapter "Material and Methods", shortened and included in the section "Introduction" (lines 53-57 in the actual manuscript). Figure 1 was removed.
Reviewer 2 Report
The study is very interesting. The paper is well written and innovative as well. Unfortunately, I have got some serious remarks, which exclude the possibilty for fast publication.
The authors mustn't analyse and judge the "musculosketelal disorders" generally. This term is too large. Before the research program they should select precise medical problems, like for example lateral epicondylitis, low back pain, frozen shoulder etc. These exampled disfunctions ougth to be analysed separately in various comparative groups. The authors should choose concrete illnesses and analyse alone. The number of analysed workers in groups is to low. The authors must return to the beginning and make a research plan one more time. I am sorry, but at this moment your conslusions do not correlate with such methodology.Author Response
Point 1: The authors mustn't analyse and judge the "musculosketelal disorders" generally. This term is too large. Before the research program they should select precise medical problems, like for example lateral epicondylitis, low back pain, frozen shoulder etc. These exampled disfunctions ougth to be analysed separately in various comparative groups. The authors should choose concrete illnesses and analyse alone.
Response 1: We agree that studying general musculoskeletal disorders might lack a precise scientific question in regard to research. However we tried to set up a clear objective for our study.
The study question was important for the health protection of workers. We wanted to examine the need for additional health promotion measures in this type of employment. There was no indication of an accumulation of a specific disease (e.g. lateral epicondylitis), which could be targeted a priori.
The problem with the selection of precise medical problems and their separate analysis in various comparative groups is that few events in a narrow indication have a lower probability of a positive test than in an expanded indication, despite the same effect (for example 1 of 2 compared to 5 of 10).
In our extensive literature search, studies were dominant which analysed individual body regions (e.g. upper extremity), but also combinations of several body regions (e.g. entire spine, upper extremity and hip/knee). The authors are of the opinion that here the focus has already been narrowed down to elbow, forearm and wrist (without shoulder and hand). Not infrequently, WMSDs are characterized by the fact that they cause non-specific complaints without a clear clinical diagnosis being detectable. This, in turn, has motivated us to stick to the common "regional" classification.
In order to optimize the focused research question, we have specified the word "musculoskeletal disorders" at the following points in the manuscript: line 2/3, 34.
Point 2: The number of analysed workers in groups is too low.
Response 2: The study was powered to prove the hypothesis regarding these study objectives, and the proof was positive. Therefore the sample size must be considered sufficient for the study question. The relatively low sample size on the other hand resulted from strictly following the European data protection legislation and from strictly avoiding the bias through economical interest on study participation.
Reviewer 3 Report
This study reports an interesting experience conducted in a workplace to evaluate the frequency of symptoms related to the involvement of the upper limb. The experimental model could be usefully applied to other companies.
The epidemiological design is accurate and is preceded by a careful analysis of the literature. The authors worked in a large company, carefully selecting cases of interest.
The final results of the research, with the frequency of symptoms almost superimposable in the exposed and control workers, testifies to the good level of industrial hygiene and work organization in the company.
Among the factors to consider in the interpretation of the results, in addition to physical workload and repetitive work tasks (Line 384), authors could consider environmental conditions [e.g.: Magnavita N, Elovainio M, De Nardis I, Heponiemi T, Bergamaschi A. Environmental discomfort and musculo-skeletal disorders. Occup Med (Lond). 2011; 61(3):196-201] and individual characteristics of the workers [Magnavita N. Work-related symptoms in indoor environments: a puzzling problem for the occupational physician. Int Arch Occup Environ Health. 2015;88(2):185-196 10.1007/s00420-014-0952-7]. These factors that influence the frequency of symptoms may explain the differences between this study and some of the literature.
It would be interesting for occupational doctors to know if workers in this company were subjected to periodic checks by a doctor. It would also be interesting to know the provisions that could be adopted in the cases of upper extremity WMSDs. Early detection of these disorders is an objective of occupational health services in the workplace.
Author Response
Point 1: Among the factors to consider in the interpretation of the results, in addition to physical workload and repetitive work tasks (Line 384), authors could consider environmental conditions [e.g.: Magnavita N, Elovainio M, De Nardis I, Heponiemi T, Bergamaschi A. Environmental discomfort and musculo-skeletal disorders. Occup Med (Lond). 2011; 61(3):196-201] and individual characteristics of the workers [Magnavita N. Work-related symptoms in indoor environments: a puzzling problem for the occupational physician. Int Arch Occup Environ Health. 2015;88(2):185-196 10.1007/s00420-014-0952-7]. These factors that influence the frequency of symptoms may explain the differences between this study and some of the literature.
Response 1: Thank you for these interesting literature references. We have incorporated them into the manuscript (lines 376-379).
Point 2: It would be interesting for occupational doctors to know if workers in this company were subjected to periodic checks by a doctor. It would also be interesting to know the provisions that could be adopted in the cases of upper extremity WMSDs. Early detection of these disorders is an objective of occupational health services in the workplace.
Response 2: Occupational health and safety is a right of the employees, as laid down in the European Union's framework directive on occupational health and safety. Since the end of 2008, occupational medical precautions in Germany have been regulated in the Ordinance on Occupational Medical Precautions (ArbMedVV).
On this legal basis, all employees in all groups of the company we examined receive occupational medical check-ups in all groups, depending on the occupational hazard, with the aim of identifying or preventing work-related illnesses. In addition, every employee is offered optional preventive medical check-ups, during which individual problems of the employee can be discussed.
In addition, a comprehensive check-up programme (duration of 3 hours) is offered in this company on a voluntary basis for all employees (i.e. all 3 groups in our examination). The primary goal of this check-up is to identify risk factors, precursors or diseases and to prevent or treat them in a targeted manner. One of the core ideas of this check-up is to fill the extensive health programs offered with the right target group. The participation rate in a prevention program after check-up is currently over 90%. Prior to the introduction of check-ups, the "wrong" target group usually participated in the generally offered prevention measures.
In the case of optional preventive measures, examinations are carried out on the basis of the complaints reported. At the check-up programme, a full body profile is prepared (patient history, blood values, ECG, lung function test, visual and hearing test, full body examination including organs, joints, etc.).
The optional preventive medical check-up can be offered at any time if there are problems on the part of the employee. The examination interval at the check-up programme is staggered according to age. Age <40 every 5 years, <50 every 3 years, > 50 every 2 years.
If abnormalities are identified during the check-ups, further targeted measures can be initiated. In the case of identified upper extremity WMSDs, hand therapeutic measures, ergonomic optimisation of the workplace and examinations by specialists (e.g. upper extremity surgeon) are usually initiated.
Over the past few years, risk factors for the development of further diseases have been identified in 80% of the workforce and initial diagnoses requiring treatment in 10%.
In addition, the Aesculap company offers a comprehensive range of preventative measures such as nutrition counseling, gym training, courses (cardio, back...), relaxation offers, cancer prevention (especially skin, colon cancer...) and individual preventive training with a sport therapist based on the problems identified during the check-up.
The authors have included the most important of the above points in condensed form in the text (lines 389-397).
Round 2
Reviewer 2 Report
I am really sorry, but the authors' reply is not enough. The experiment (not paper) needs improvements (my previous report). The topic is interesting, and in future it could be a great article.
I do not recommend this manuscript for publication, so the editors must make the final decision.